# Scalability of a singing-based intervention for postpartum depression in Denmark and Romania: protocol for a single-arm feasibility study

Katey Warran [ORCID],[1] Calum Smith,[2,3] Hanna Ugron,[4] Louise Frøkjær Carstens,[5] Rarita Zbranca,[4] Mikkel Ottow,[6] Oana Maria Blaga,[7] Nicolai Lund Ladegaard,[8] Rachel E Davis [ORCID],[9] Daisy Fancourt [ORCID],[1] Nils Fietje [ORCID][2]

KW and CS are joint first authors.

## ABSTRACT

**Introduction** Postpartum depression (PPD) affects around one in seven women globally, with these women in need of non-pharmaceutical treatment strategies. There is a long history of the benefits of singing for maternal mental health, and promising research exists showing the clinical effectiveness of group singing. Group singing interventions are being scaled up to support new mothers in the United Kingdom, but we do not know if such an intervention may benefit women in different cultural contexts. This protocol focuses on exploring the feasibility of implementation and perceived impact of a 10-week group singing intervention for new mothers in Romania and Denmark eliciting signs of PPD.

**Methods and analysis** Data will be collected from up to 48 women with a score ≥10 on the Edinburgh Postnatal Depression Scale (EPDS) participating in a 10-week group singing intervention in Denmark or Romania, as well as a range of project stakeholders. The singing classes will take place in person and be facilitated by professional singing leaders. Feasibility of implementation will be analysed through qualitative data (eg, focus groups, interviews) and quantitative data (eg, the Feasibility of Intervention Measure). Perceived impact will be explored via surveys that include mental health measures (EPDS, Multidimensional Scale of Perceived Social Support, WHO Five Well-Being Index) from singing intervention participants (at weeks 1, 6, 10) and focus groups. Descriptive statistics, repeated measures analysis of variance and analysis of covariance will be used to analyse quantitative data. Framework method and thematic analysis will be used to analyse qualitative data.

**Ethics and dissemination** The national ethics committees in Romania (IRB-PH Protocol #2021-211217-012) and Denmark (case number 1-10-72-274-21) have approved the study, as has the Ethics Review Committee at the World Health Organization (ERC.0003714). All participants will be required to provide informed consent. Results will be disseminated by reports published by the WHO Regional Office for Europe, peer-reviewed publications and at conferences.

## INTRODUCTION

Postpartum (also called postnatal) depression (PPD) is a type of depression that can occur

For numbered affiliations see end of article.

**Correspondence to**
Dr Katey Warran;
k.warran@ucl.ac.uk

### STRENGTHS AND LIMITATIONS OF THIS STUDY

⇒ Comprehensive range of implementation measures collected to explore feasibility of implementation.

⇒ Uses mixed-methods to explore the implementation of a complex singing intervention from multiple perspectives.

⇒ The study does not include a control group; therefore, it will not be possible to make generalisable claims in relation to the quantitative impact of the intervention.

⇒ Includes data collection from mothers and study managers/partners, strategic managers/partners and referrers, to provide broad perspectives on the feasibility of implementation.

⇒ Includes an embedded qualitative component to explore the subjective impact of the intervention.

during the first year of having a baby.[1] PPD affects around one in seven women globally,[2] and it can have significant impact on mental health (and consequently mental health services), even developing into suicide if left untreated.[1 3 4] Symptoms include depressive mood, reduced interest in daily activities, insomnia, agitation, guilt and fatigue, all of which can also affect a mother's relationship with her child.[2] Notably, it has been suggested that depressed mothers exhibit two interaction styles of being 'withdrawn' or 'intrusive', which may adversely affect babies due to inadequate stimulation and arousal modulation.[5] In the longer term, this can also adversely affect a child's behaviour[6] and academic performance.[7] Therefore, identifying ways to deliver holistic healthcare that can provide psychosocial support is particularly important for this group.

### Treatment for PPD

There is no complete treatment solution for PPD. Pharmacological treatment shows

promise (eg, brexanolone) but uptake and adherence are low, with treatment often delayed.[8 9] While psychological therapies have been shown to improve mood, the benefits are short term and not superior to spontaneous remission.[10 11] Additionally, there can be stigma attached to receiving treatment, and mothers can find it hard to make time to attend psychological therapies.[12]

In recent years, there has been a burgeoning interest in how social, arts and community group-based activities can support mental health, quality of life and well-being, alongside increased interest in finding cost-effective evidence-based interventions that can be embedded into healthcare pathways, such as via social prescribing schemes.[13–16] Group-based activities have been shown to prompt social, psychological and behavioural responses that are linked with improved mental and physical health.[17] In the context of support for new mothers, mother–baby groups that include play activities have been shown to relax mothers, improve social interaction and provide personal fulfilment, with those who join in pregnancy also displaying a reduction in depressive symptoms.[18] However, singing groups may be able to provide greater support for depressive symptoms than play activities.[19]

### Singing for maternal mental health
There is a long history of the benefits of singing for maternal mental health. In anthropology and evolutionary theory, it has been argued that lullabies and mother–infant singing evolved from motherese (a style of exaggerated speech between mother and baby), which was used to improve bonding and reassure babies.[20 21] Studies exploring regular singing between mother and baby have shown that singing can improve mother–infant bonding in the first 9 months following birth.[22] These findings provide strong grounds to explore the clinical effects of singing sessions for new mothers. In a randomised controlled trial called 'Music and Motherhood' comparing the effects of 10-week singing and play programmes for 134 mothers with symptoms of PPD, it was found that singing reduced symptoms by 38% and led to a significantly faster improvement in symptoms for those with moderate–severe PPD symptoms. As part of the same trial, qualitative research also supported these findings, showing that reduced symptoms were facilitated through fostering a functional emotional response rooted in the needs of new motherhood, including providing 'me time' for mothers and 'tools' for calming and bonding with babies.[23]

### Music and Motherhood and the SHAPER Study
The 'Music and Motherhood' trial comparing the effects of singing and play interventions for PPD was led from 2015 to 2017 by the Centre for Performance Science (a joint research centre that sits between the Royal College of Music and Imperial College, London) with support from University College London (UCL) in the United Kingdom.[24] Following the research, the programme was developed into practice for the first time and expanded from 2017 onwards by an organisation called Breathe Arts Health Research as part of their 'Melodies for Mums' Programme. The team embedded the intervention in support services in South London (Lambeth and Southwark) and later nationally as an online intervention during the COVID-19 pandemic. In order to explore the effectiveness of the intervention on a larger scale in London, the 4-year 'Scaling-Up Health-Arts Programmes: Implementation and Effectiveness Research-Postnatal Depression' (SHAPER-PND) Study was launched in 2020 and is ongoing.[25]

### The importance of culturally sensitive psychosocial interventions
Although the SHAPER-PND Study is looking to explore the impact of Music and Motherhood on a larger scale, it is limited to the United Kingdom. We therefore do not know if this intervention may benefit women in different cultural and linguistic contexts. Given the complexity of arts interventions as moderated by their local environments and involving multiple components,[17 26] the first step to implementing such an intervention in a new setting is understanding how it might need to change to meet local needs. Gaining this understanding requires a level of cultural competence to frame information and deliver interventions that are sensitive to cultural values, beliefs and attitudes towards mental health conditions.[27] Therefore, there is an important gap in the literature regarding researching how an intervention needs to be changed to meet health challenges in contexts beyond which it may have been initially designed.

### This protocol
This feasibility study explores how to implement the Music and Motherhood intervention in Denmark and Romania. Both of these countries present a strong rationale for implementation.

In Romania, data on PPD are scarce and mental health literacy of PPD is poor.[28] For example, out of 112 pregnant women in their third trimester interviewed in clinical settings in 2019, 52% reported not knowing what PPD is, whereas 43% considered that they could never develop PPD.[29] The limited available research shows that up to 24% of new mothers could be affected by PPD.[30] The prevalence of PPD in Denmark has been found to be between 5.5%[31] and 7.5%.[32] National reports on available interventions targeting PPD in Denmark stress the need for early and sensitive care and services. In particular, it has been highlighted that community group-based interventions can reduce feelings of shame and increase empowerment.[33] However, there is little research on the benefits of group singing interventions for this population.

Our primary aim is to evaluate the feasibility of implementation in these new cultural contexts, and our secondary aim is to evaluate the perceived impact of the intervention on mental health and well-being for intervention participants. The study is funded by the Nordic

Culture Fund (project number 24 479), with further support from the Wellcome Trust (219425/Z/19/Z), Region Midtjylland (Denmark) and the Cluj Culture Centre (Romania), and delivered by Dr Nils Fietje at the WHO Regional Office for Europe (sponsor) in partnership with UCL.

## METHODS AND ANALYSIS

### Setting

The singing intervention will be implemented in community centres in Denmark and Romania. The study is being conducted in partnership with Den Kreative Skole in Region Midtjylland (Central Denmark Region) in Denmark, and Cluj Culture Centre in Cluj, Transylvania in Romania. Staff from these organisations will lead the delivery, data collection and analyses for the study in their respective countries.

There will be two 10-week singing programmes in each country, delivered by professional singing leaders. The implementation team and singing leaders will receive international training from Breathe Arts Health Research. The spaces within which the intervention will be run will have suitable space for 8–12 prams, sufficient baby changing facilities and will be easily accessible using public transport. The locations will not be disclosed in recruitment and marketing due to the potential stigma attached to having PPD. In Romania, due to the low uptake of COVID-19 vaccination, the intervention will also be delivered in a semioutdoor setting to ensure good ventilation. In both countries, participants will be asked to take a rapid antigen test ahead of attending each week, and tests will be supplied at no cost to the participants, if needed.

### Patient and public involvement

This protocol was developed and discussed with two patient and public involvement (PPI) groups—one in Denmark and one in Romania. The PPI groups consist of mothers with lived experience, academic experts and mental health professionals. The groups plan to meet a further two times throughout the study to guide the project's delivery, including supporting the development of musical content for the singing sessions.

### Participants

Data will be collected from different project stakeholders (coined the 'implementation team' for this protocol) to examine how the intervention is delivered. This will include four groups:

1. Strategic managers/partners (n=7) involved in planning and overseeing the intervention delivery and success (eg, members of the management committee from Region Midtjylland, Cluj Culture Centre and Den Creative Skole who meet regularly with the WHO Regional Office for Europe and UCL for planning meetings).

2. Project managers/partners (n=7) involved in coordinating and managing the on-the-ground delivery of the intervention (eg, project coordinators working at Cluj Culture Centre and Region Midtjylland).
3. Referrers (n≈8) (such as health nurses in Denmark or other health/non-governmental organisation (NGO) professionals in Romania) who refer participants to be included in the intervention.
4. Singing leaders (n=3) who will lead the practical and creative delivery of the intervention. One in Denmark, two in Romania.

All managers/partners and singing leaders will need to speak English sufficiently enough to read the information sheet and give informed consent (see online supplemental files 1 and 2). Referrers and singing leaders will be able to participate in Danish or Romanian. All stakeholders must be aged over 18 years to participate. Project managers/partners from across both countries will engage in regular biweekly online meetings with the strategic management team as a form of supervision, ensuring ongoing reflection on ethical practice (in relation to both delivery of the intervention to participants and ensuring staff well-being) and fidelity of delivery.

Data will also be collected from our singing intervention participants to explore the perceived impact of the singing classes. Based on class sizes that have been successful in the context of the United Kingdom,[19] the aim is to include two groups of 8–12 participants in each country, a total of 32–48 participants in total across both Denmark and Romania. Of note, we did not calculate our sample size in view of determining statistical power, instead focusing on how to feasibly implement the intervention and explore its perceived impact. The inclusion criteria are women who:

► Score ≥10 on the Edinburgh Postnatal Depression Scale (collected upon screening).
► Are up to 40 weeks post-birth with an infant aged 0–9 months.
► Are aged 18 years or over.
► Have the capacity to give informed consent.

Potential singing intervention participants who cannot speak Danish, Romanian, Hungarian or English to a level that is sufficient to read and understand the information sheets and consent forms for the study (either through reading or being read to) will be excluded, as will those under the age of 18 years or those who have infants not aged 0–9 months (see participant consent forms in online supplemental files 3 and 4). Participants can still join if they are receiving other treatments.

### Recruitment

#### Implementation team

All stakeholders involved in the delivery and running of the intervention will be invited to participate. Strategic and project managers/partners who are already involved in the delivery of the project will be sent an email outlining what participation in the study will involve. Referrers will be approached within countries by

on-the-ground teams either in person or digitally, where the study will be explained, and they will have the option to join the research. The singing leaders will be informed of the research when they are employed to lead the classes and asked to provide informed consent for their participation. They will have the right to withdraw from the study, but it will be discussed on a case-by-case basis regarding whether ending participation in the research would also end their employment, based on what stage of the study they choose to withdraw (ie, reflecting on if there is time to recruit an alternative lead to participate in the research).

### Denmark intervention participants

The recruitment strategy in Denmark will follow a referral process whereby health nurses in the municipal health authority in Central Denmark Region will screen for eligibility for the intervention and, if consented, participants will be referred directly into the singing intervention. We will also supplement this approach with reaching out to relevant local NGOs and mother and baby groups (online and in person) to ensure a diverse sample. This recruitment process started in spring 2022 and ended in autumn 2022.

### Romania intervention participants

As there is no structure in place to enable direct referral vial healthcare services, the recruitment strategy in Romania will centre around printed and digital marketing and communications activities from the Cluj Cultural Centre. This will include dissemination of printed materials through community facilitators, adverts on local radio stations and social media. The team will also contact gynaecological clinics (private and public), family physicians and NGOs that support mothers after they give birth to share information about the project. In addition to circulating printed and digital materials, project team members will attend relevant groups and clinics (with permission from relevant stakeholders) to speak to potential participants face-to-face about the project. This recruitment process started in spring 2022 and ended in early summer 2022.

In both countries, we will ensure that our recruitment approach is one founded upon values of inclusivity and diversity, seeking to ensure that all those who wish to engage can. Participants will be consented by the project manager in each respective country. For participants who request to join the intervention, but who are not eligible (see inclusion criteria), we will aim to offer one free singing workshop session as a recognition of their interest in being involved and be referred to other interventions and/or activities that may be of interest (eg, given a list of free arts programmes, local choir groups).

### The singing intervention

The starting point for the design and structure of the singing sessions was the Music and Motherhood intervention.[24] Informed by the model from the United

Kingdom, it is anticipated that the classes will happen on a weekly basis for 8 weeks, last 1 hour and include mothers sitting in a circle with their babies to learn songs together. There will be time to socialise before and after the class, and it is expected that the group will learn culturally diverse songs and sometimes use small instruments.[24] However, a key aim of our study is to explore how this (United Kingdom) intervention may need to change to meet local needs. The study teams will work with our PPI groups to tailor the intervention to new contexts and receive guidance and advice from Breathe Arts Health Research (United Kingdom) in the form of implementation team and singing lead training sessions. The singing leader will bring their own ideas for content to discuss with the PPI group, and be trained to respond to participants' needs throughout the study, documenting any changes made so that it is possible to understand how and why the intervention changed throughout processes of delivery.

From current feedback from PPI groups in Denmark and Romania, it is expected that songs included in the classes will cover a range of genres and will not be tied to a specific language, but rather feature phonetic noises and percussive sounds. This is to ensure that, even in groups involving mothers from minority groups (Hungarian, Roma mothers), who do not share the same first language with the other Romanian mothers, no individuals will be disadvantaged. In addition, we will have a Hungarian and a Romanian-speaking music leader, to reduce any language barrier.

The classes will run from late spring to the end of 2022. The two classes in each country will have some overlap, with the first tranche of mothers beginning their participation in the intervention slightly earlier than the second tranche. This is to ensure that all the classes can take place outside in warm months if they need to due to high incidence of COVID-19.

### Data collection

#### Feasibility of implementation methods

Table 1 shows an overview of the data collection methods used to meet our primary aim of evaluating the feasibility of implementation, with the key groups and individuals who will be approached listed in the columns.

These implementation measures are informed by frameworks from implementation science including:

► The Reach Effectiveness Adoption Implementation Maintenance (RE-AIM) framework.[34]
► Proctor *et al*'s (2011) outcomes for implementation research taxonomy.[35]
► The Medical Research Council's guidance on carrying out process evaluations of complex interventions.[36]
► Warran *et al*'s (2022) Ingredients iN ArTs in hEalth (INNATE) framework.[26]

The data collection methods informed by these frameworks are described in more detail below.

**Table 1** Data collection methods to evaluate the feasibility of implementation of Music and Motherhood

| Data collection method* (described in detail below) | Strategic managers/ partners | Project managers/ partners | Referrers | Singing leader | Intervention participants |
|---|---|---|---|---|---|
| Implementation team focus group | X | x | x | | |
| Implementation team short survey | X | x | x | x | |
| Implementation team semistructured interview | | | | x | x |
| Track record document | X | | | | |
| Screening and attendance log | | x | | x | |
| Active ingredients worksheet | | | | x | |

*Topic guides informed by Soukup *et al* (AIM)[43] and the Scaling-up Health-Arts Programmes: Implementation and Effectiveness Research implementation protocols (one-to-one interviews and focus groups).[25]
AIM, Acceptability of Intervention Measure.

## Implementation team focus group

Implementation focus groups will be conducted with three different groups (one focus group for each group): strategic managers/partners, project managers/partners and referrers. The topic guide for the group will ask questions to assess the following about the intervention: acceptability, appropriateness, feasibility, adoption, sustainability, unintended consequences, implementation strategies, contextual factors, success of any training delivered and key learning points from the project. The focus groups will take place post-intervention and are expected to last 60–90 min.

## Implementation team short survey

A short survey containing three validated measures of four items each (Acceptability of Intervention Measure (AIM), Intervention Appropriateness Measure (IAM) and Feasibility of Intervention Measure (FIM)) will be distributed to strategic managers/partners, project managers/partners, referrers and the singing leader to complete post-intervention. The survey is anticipated to take approximately 5 min.

## Implementation one-to-one interview

An interview will be carried out with the singing leader and with a selection of participants (n=4–5, the first volunteers will be selected) who have taken part in the intervention. We will also invite those who drop out to participate in a one-to-one interview if they would like to. The topic guide for the interviews will ask questions to assess the following: acceptability, appropriateness, feasibility, fidelity of receipt (participants only), adoption, sustainability (singing leader only), unintended consequences, contextual factors (singing leader only), adequacy of training delivered (singing leader only) and sense of preparedness (singing leader only). The interviews will take place post-intervention and are expected to last 60 min each.

## Track record document

A document is being created and maintained by the WHO Regional Office for Europe to record meeting minutes of management committee meetings (with strategic managers/partners) and meetings with partners in Denmark and Romania. This document will serve as a resource to document our implementation processes, how decisions were made throughout and to explore key learning points from the project, as well as to assess fidelity of delivery.

## Screening and attendance logs

Logs will be created and maintained by project managers/coordinators to document the number of people who were screened and how many of these individuals went on to participate in the intervention. A record will also be kept of intervention attendance by the singing leader, including whether any participants drop out throughout the 10-week programme. This will help us to assess the reach of the project (ie, whether the intended audiences came into contact with the intervention and how).

## Active ingredients worksheet

To aid the process of tracking and understanding any adaptations made to the intervention throughout the process of implementation, the INNATE framework will be used.[37] This framework puts forward a comprehensive mapping of 139 active ingredients (the components of an intervention that may prompt the mechanisms of action underlying potential health outcomes) which can be used to describe, in depth, the key aspects of an intervention that may be important to health. The framework will be used to identify, and reflect on, intervention active ingredients at the planning stage of the intervention (ie, the week before the classes are implemented in each country) and then again after the intervention is complete (ie, after week 10). This tool will therefore support with assessing fidelity of delivery through documenting adaptations to the intervention that occur organically, as well as provide a document that could be used to support with replication of the intervention into other countries in the future.

**Table 2** Quantitative measures to evaluate the perceived impact of the intervention on mental health and well-being for singing participants

| Outcome | Type | Measure | When |
|---|---|---|---|
| Postpartum depression | Quantitative | Edinburgh Postnatal Depression Scale | Weeks 1, 6 and 10 |
| Social support | Quantitative | Multidimensional Scale of Perceived Social Support | Weeks 1, 6 and 10 |
| Well-being | Quantitative | The World Health Organization Five Well-Being Index | Weeks 1, 6 and 10 |

## Perceived impact of intervention methods

### Survey

Intervention participants will be invited to complete a short (5-minute) quantitative survey at three time points throughout study participation (weeks 1, 6 and 10). Participants will be contacted via email with the surveys and followed up with email reminders to improve adherence. These surveys have been kept purposely short to try and keep the process simple and easy for participants to engage in (see table 2).

### Focus group

All intervention participants will be invited to join one focus group at the end of the 10-week programme (one focus group per singing group). The groups will explore subjective experiences of the weekly sessions such as in relation to what participants enjoyed/did not enjoy, any challenges that occurred, and the perceived impact of the classes on PPD symptoms, social relationships, and identity. The focus group is anticipated to last 60–90 min. Given the potential sensitive nature of discussing maternal mental health, focus group facilitators will receive specific training delivered by UCL on how to conduct these sessions in a sensitive way.

### Optional reflective journal entry

Study participants will be given the option to complete a journal entry at the end of the singing programme to reflect on anything they feel they have been unable to share at other data collection points.

In addition to the above qualitative and quantitative data, we will also collect sociodemographic data (location, age, ethnicity, educational attainment).

See figure 1 for a flow diagram of all of our data collection time points and measures.

## Data analyses

### Quantitative

Data will be analysed using Stata and/or MASPSS. Descriptive statistics will be used to analyse the implementation short survey (AIM, IAM, FIM), and repeated measures analysis of variance will be used to explore changes to mental health and well-being across time for our singing intervention participants (within-subject changes). Analysis of covariance will additionally explore whether these changes are moderated by participants' sociodemographic backgrounds.

### Qualitative

Framework method will be used to organise and analyse the qualitative data collected from intervention participants relating to implementation (ie, one-on-one interviews).[38] The study teams across Denmark and Romania, with support from UCL, will develop a framework for use that is underpinned by approaches from implementation science, as previously outlined (eg, RE-AIM, Proctor *et*

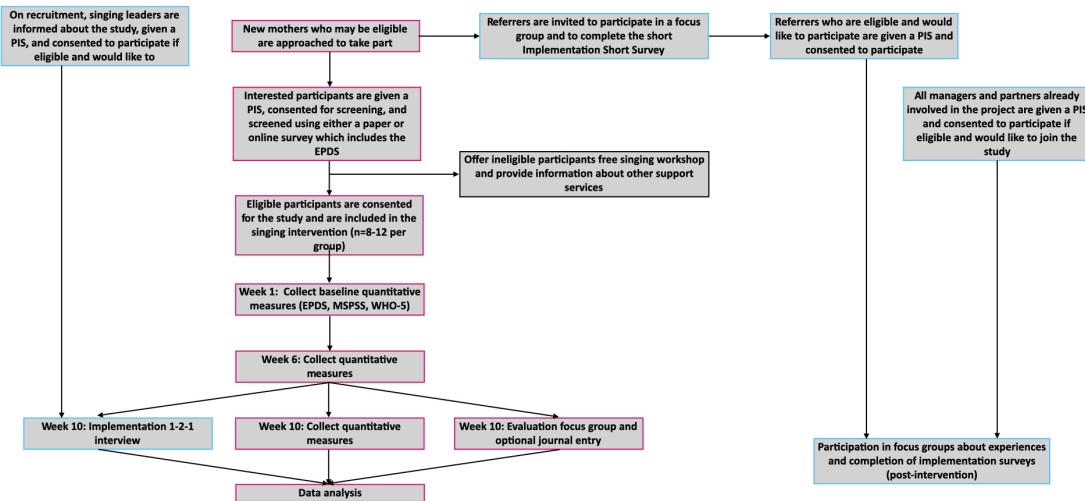

**Figure 1** Study flow chart. Diagram showing data collection time points and measures collected for singing intervention participants and the implementation team. EPDS, Edinburgh Postnatal Depression Scale; MSPSS, Multidimensional Scale of Perceived Social Support; PIS, participant information sheet; WHO-5, WHO Five Well-Being Index.

*al*'s taxonomy). The framework will be used to construct qualitative themes from the data that are relevant to our primary aim of evaluating implementation.

Drawing upon guidance from Braun *et al,*[39–41] a reflexive thematic approach will be taken to analyse and evaluate experiences of the singing group and its perceived impact (ie, from the focus groups and journal entries). This will involve generating initial codes, searching for and reviewing themes, and then defining and naming themes. An inductive approach will be taken, working from the bottom-up, generating or drawing out knowledge from the data through examination of what is meaningful for the participants, thereby placing participant experiences at the heart of the analysis.[42] This will allow us to evaluate perceived changes to mental health and well-being. Qualitative analysis software will be used to analyse and organise qualitative data.

Analyses will be undertaken by researchers in each country who were not part of the intervention development or any management processes. Project managers, researchers and members of the management team will also engage in regular biweekly meetings throughout data collection and analyses to engage in discussion about emerging findings as a form of reflexivity on our processes, thereby improving the rigour of the study.

## ETHICS AND DISSEMINATION
### Ethics
The National Ethics Committees in Romania (IRB-PH Protocol #2021-211217-012) and Denmark (case number 1-10-72-274-21) have approved the study, as has the Ethics Review Committee at the World Health Organization (ERC.0003714). All participants will need to provide informed consent in order to participate in the study. Consent forms for both countries are included as online supplemental files.

### Dissemination
The findings will initially be written up in a project implementation report, published by the WHO Regional Office for Europe and made publicly available. The aim of this report will be to share our processes and learning with commissioners, relevant organisations, policymakers and healthcare institutions so that it can be used in the future to provide guidance and best practice related to how arts and health interventions can be implemented in and scaled to different national and cultural contexts. Findings will also be published in peer-reviewed journals and shared at conferences, meetings and events, including webinars run by the WHO Regional Office for Europe.

### Data protection and confidentiality
In Denmark, all data will be stored and handled following the Danish law, specifically databeskyttelsesloven and databeskyttelsesforordningen. In Romania, data will be stored in line with local data management laws. In both countries, consent forms and identifiable data will be stored securely and only accessible by the project manager in each country. The quantitative data will be anonymised and the qualitative transcripts will be pseudonymised before being shared with researchers in each respective country for analysis. All identifiable data will be stored separately to participant demographics. Quantitative data will be collected using REDCap in Denmark and using Microsoft Teams in Romania. Any extracted data used for analysis will be stored on secure, password-protected storage systems that only the researchers doing the analysis and the project managers will have access to. Audio recordings from qualitative interviews will be deleted immediately from hand-held devices and from storage facilities as soon as transcripts have been created. All identiable information will be deleted at the end of the study. Anonymised and pseudonymised data will be stored in line with data collection policies in each country—5 years in Denmark and 5 years in Romania. Any adverse events or unintended consequences will be discussed at management meetings and documented by NF and CS at WHO/Europe in the track record document (see the Track record document section).

**Author affiliations**
[1]Research Department of Behavioural Science and Health, University College London, London, UK
[2]Behavioural and Cultural Insights Unit, WHO Regional Office for Europe, Copenhagen, Denmark
[3]Nuffield Department of Population Health, University of Oxford, Oxford, UK
[4]Centrul Cultural Clujean, Cluj-Napoca, Romania
[5]Den Kreative Skole, Region Midtjylland, Silkeborg, Denmark
[6]Region Midtjylland, Viborg, Denmark
[7]Center for Health Policy and Public Health, Babeş-Bolyai University, Cluj-Napoca, Romania
[8]Department of Clinical Medicine, Aarhus University, Aarhus, Denmark
[9]Centre for Implementation Science, King's College London, London, UK

**Acknowledgements** The authors would like to thank Dr Rosie Perkins (Royal College of Music) and Dr Jill Sonke (University of Florida) for their peer review of the WHO Regional Office for Europe ethics application that informed this protocol. They would also like to thank Yvonne Farquharson, Hannah Dye and Lorna Greenwood from Breathe Arts Health Research for the training provided and insight shared in view of their Breathe Melodies for Mums Programme.

**Contributors** All authors listed made contributions to the conception and design of the study protocol. CS and KW equally led the drafting of the protocol report for this publication, with specific feedback from NF. KW led the design of the methodology for the data collection, and NF led the overall supervision of the project, including funding acquisition. HU and RZ provided specific advice on the Romanian context, and LFC and MO provided advice on the Danish context. NLL and OMB supported with developing data protection and data handling procedures. RED provided insights on the implementation measures based on her experience of working on the SHAPER Study (United Kingdom), and DF provided extra supervisory support in view of developing the United Kingdom intervention. All authors have approved the final protocol for publication.

**Funding** This work is supported by the Nordic Culture Fund (project number 24479), with further support from the Wellcome Trust (219425/Z/19/Z), Region Midtjylland (Denmark) and the Cluj Culture Centre (Romania).

**Disclaimer** The Nordic Culture Fund that funded the study has played no part in the development of this protocol.

**Competing interests** None declared.

**Patient and public involvement** Patients and/or the public were involved in the design, or conduct, or reporting, or dissemination plans of this research. Refer to the Methods section for further details.

**Author note** Disclaimer: The authors affiliated with the World Health Organization (WHO) are alone responsible for the views expressed in this publication and they do not necessarily represent the decisions or policies of the WHO.

**ORCID iDs**
Katey Warran http://orcid.org/0000-0002-8962-3575
Rachel E Davis http://orcid.org/0000-0003-2406-7181
Daisy Fancourt http://orcid.org/0000-0002-6952-334X
Nils Fietje http://orcid.org/0000-0002-6901-0771

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
