## [Reviewer comments · BMJ Open]

ARTICLE DETAILS

TITLE (PROVISIONAL)	Scalability of a singing-based intervention for postpartum depression in Denmark and Romania: protocol for a single-arm feasibility study
AUTHORS	Warran, Katey; Smith, Calum; Ugron, Hanna; Frøkjær Carstens, Louise; Zbranca, Rarita; Ottow, Mikkel; Blaga, Oana; Lund Ladegaard, Nicolai; Davis, Rachel; Fancourt, Daisy; Fietje, Nils

VERSION 1 – REVIEW

REVIEWER	Melissa Forbes University of Southern Queensland, Centre for Heritage and Culture
REVIEW RETURNED	23-Aug-2022

GENERAL COMMENTS	Thank you for the opportunity to review this protocol. The authors propose to scale up a group singing intervention for mothers with postnatal depression. Building on successful implementation in the UK, the scaling up extends the intervention to participants with postnatal depression in Romania and Denmark, which will provide the opportunity to gather data on the group singing intervention in two new and different cultural contexts. Staff from organisations in Denmark and Romania will lead delivery, data collection and analyses. Sessions will be led by professional singers. The patient and public involvement in the design of the protocol is commendable and it was heartening to read later that experienced singing group facilitator/s will have input into the PPI group. The practical experience of leaders will be important in understanding if and how the intervention needs to be customised to suit different cultural contexts. Consideration has been given to participant inclusivity in the Romanian context, and presumably the same broad principles of inclusivity apply in the Danish context? There is evidence of good feasibility of delivery of the group singing intervention (with some finessing to suit new cultural contexts). The implementation measures draw on existing comprehensive implementation frameworks. In regards to interviewing singing group facilitators, consider seeking input on adequacy of training and sense of preparedness to undertake group facilitation in this context, as there is currently a paucity of research on this (including skills required, skills gaps etc). Table 1 should indicate the singing group leader will take attendance logs (currently says participants, but this must be an error?).
---

	Recruitment: The protocol states that singing leaders will be informed of the research when they are employed to lead the class. Does this mean that employment is conditional upon research participation? This would need to be stated explicitly prior to any formal employment agreement being entered into. Similarly, would employment end if the participant sought to withdraw from the research? I am unable to comment on the appropriateness of the proposed methods of quantitative data analysis. Regarding analysis of qualitative data, the use of thematic analysis, a theoretically flexible analytic method, is appropriate to the stated desire to place participants' experience at the heart of the analysis. Note Braun and Clarke's new comprehensive text on RTA published in 2021 as an updated reference. I very much look forward to reading the results from this important study. line 42 of the abstract contains two typos - particualr; mother's (no apostrophe)
--	---

REVIEWER	Gunter Kreutz Carl von Ossietzky Universitat Oldenburg Institut fur Musik, Music
REVIEW RETURNED	02-Sep-2022

GENERAL COMMENTS	The authors have composed a protocol for a single-armed trial on postpartem depression. I believe that the protocol is promising and could lead to a timely and important study to address a major concern in a highly vulnerable target group that suffers from mental health issues. For this reason alone, the study should be worthy to be conducted. Along the same line, optimizing the protocol will be crucial to perform the study in a convincing manner. General issues The protocol is ambitious, well thought-through, but with shortcomings as will be explicated below. In my view there are a number of general issues to be addressed:  1. Data protection strategies are not being discussed. There will be a mass of confidential information in relation to potential mental health issues in the group. Managing this information, securing data protection, confidentiality etc. seems to be reflected very little. Who will be eligible to read protocols and for what purpose? What will informed consent entail and how will it be managed? Approvals of Ethic committees in Denmark and Romania notwithstanding, some of the guidelines to ensure a secured information flow should be reflected in the manuscript. 2. In my view, to achieve a high methodological standard, an identification and separation of assessment and intervention teams for the purpose of blinding seems necessary. As it stands, the protocol looks like the entire team will look into any data at any time. However, that will preclude blinding and, implicitly, undermine the objectivity of the research process. 3. I applaud to the manifold efforts to secure feasibility and in-depth understanding of the implications of the intervention. Nevertheless,
---

	the mixed-methods approach makes it rather hard to discern how the N for the quantitative part is estimated, perhaps based on expected effect sizes. However, there seems to be little information about expected effect sizes. 4. I wonder how mental health issues could be subject to focus group interviews given the sensitive nature of the subject matter and the exposure of intimate and potentially stressful information in a group setting with strangers. I am sure that the team leaders may have experience in that but I am still not sure how this can work. 5. Given that the interventionists work with vulnerable people, please highlight the plans for supervision and securing the interventionists' wellbeing. And how are they trained to ensure that they understand the protocol and will be ready to comply to treatment fidelity measures. 6. How are data managed? Will the team use systems such as Redcap or thelike? Minor issues The title says "scalability", but the aims state "feasibility study". The Abstract talks about group singing being "scaled up", which is unclear to me. So, what is it then? Otherwise scalability seems to play no role in the description. Abstract "Although Postpartum depression (PPD) affects around one in seven women globally, there is currently no complete treatment solution." I am struggling with this line. What does it mean? Trying to be more constructive, I suggest to say: "Postpartum depression (PPD) affects around one in seven women globally. These women are in need for non-pharmaceutical treatment strategies." - Perhaps something like that would be more appropriate.
--	---

VERSION 1 – AUTHOR RESPONSE

Reviewer: 1

	Reviewer 1	Our response
8	Consideration has been given to participant inclusivity in the Romanian context, and presumably the same broad principles of inclusivity apply in the Danish context?	Thank you for noting this important addition to our manuscript. Yes, we have indeed sought to be as inclusive as we can in our recruitment approach. We have added that we have supplemented our recruitment approach in Denmark with reaching out to local NGOs and mother and baby groups, as well as stated that our approach is underpinned by values of inclusivity and diversity in both countries.
9	There is evidence of good feasibility of delivery of the group singing intervention (with some finessing to suit new cultural contexts). The implementation measures draw on existing comprehensive implementation frameworks. In regards to interviewing singing group facilitators, consider seeking input on adequacy of training and sense of preparedness to undertake group facilitation in this context, as	Thank you to the reviewer for noting that we have good feasibility of delivery of the group singing intervention. We also agree that there is a lack of research exploring the training needs of singing leaders and their preparedness to work in contexts such as this. We will be asking questions on training and have added that we will also ask about sense of preparedness. We have also

	their is currently a paucity of research on this (including skills required, skills gaps etc). Table 1 should indicate the singing group leader will take attendance logs (currently says participants, but this must be an error?).	corrected the error in the table with regards to collecting attendance.
10	Recruitment: The protocol states that singing leaders will be informed of the research when they are employed to lead the class. Does this mean that employment is conditional upon research participation? This would need to be stated explicitly prior to any formal employment agreement being entered into. Similarly, would employment end if the participant sought to withdraw from the research?	Thank you for noting this important ethical consideration. We are collecting informed consent from singing leaders upon employment and they will have the right to withdraw to ensure ethical practice. However, it would depend on when they withdraw as to whether this would impact upon the length of their contract. If they were to withdraw in the very early stages, we would seek an alternative singing leader in the hope that we could also collect in-depth data on this aspect of the study. If it were in the later stages and not feasible to do this, we would not collect data from the singing leader. However, this would be discussed on a case-by-case basis, to ensure we could learn about the role of the singing leader in implementing this intervention as best we can, whilst also ensuring good ethical practice.
11	Regarding analysis of qualitative data, the use of thematic analysis, a theoretically flexible analytic method, is appropriate to the stated desire to place participants' experience at the heart of the analysis. Note Braun and Clarke's new comprehensive text on RTA published in 2021 as an updated reference.	Thank you very much for this suggestion – a great addition to ensure our qualitative analyses is informed by the latest approach developed by Braun and Clarke.
12	line 42 of the abstract contains two typos - particualr; mother's (no apostrophe)	Thank you for noticing this. We have now updated the text.

Reviewer: 2

	Reviewer comments	
13	1. Data protection strategies are not being discussed. There will be a mass of confidential information in relation to potential mental health issues in the group. Managing this information, securing data protection, confidentiality etc. seems to be reflected very little. Who will be eligible to read protocols and for what purpose? What will informed consent entail and how will it be managed? Approvals of Ethic committes in Denmark and Romania notwithstanding, some of the guidelines to ensure a secured information flow should be reflected in the manuscript.	Thank you very much for noting that we need to address data protection strategies in our protocol. We have now added a section entitled 'data protection' in the 'Ethics and dissemination' section of the manuscript. Here we outline that we will ensure data is stored in line with local data management laws, as well as describe who will have access to identifiable information and consent forms and where data will be securely stored and deleted.
14	2. In my view, to achieve a high methodological standard, an identification and separation of assessment and intervention teams for the purpose of blinding seems necessary. As it stands, the protocol looks like the entire team will look into any data at any time. However,	This is certainly something we are mindful of and we are ensuring that the people collecting and analysing data on-the-ground are not the same people in our management team. We are also engaging in regular, collective reflexive processes. We have now added this

	that will preclude blinding and, implicitly, undermine the objectivity of the research process.	extra detail into the manuscript.
15	3. I applaud to the manifold efforts to secure feasibility and in-depth understanding of the implications of the intervention. Nevertheless, the mixed-methods approach makes it rather hard to discern how the N for the quantitative part is estimated, perhaps based on expected effect sizes. However, there seems to be little information about expected effect sizes.	Thank you for highlighting that we need to explain how we determined our sample size. As noted to reviewer 1, we decided our singing class sizes based on what has been successful in UK contexts. We did not calculate our sample size based on expected effect sizes. We have included a small number of quantitative validated scales in order to determine perceived impact (combined with our qualitative focus groups). As this is a small-scale study with no control group focused primarily on feasibility of implementation with an embedded mixed-methods component to explore subjective impact, we did not seek to determine a quantitative association between singing and reduced post-natal depression.
16	4. I wonder how mental health issues could be subject to focus group interviews given the sensitive nature of the subject matter and the exposure of intimate and potentially stressful information in a group setting with strangers. I am sure that the team leaders may have experience in that but I am still not sure how this can work.	At UCL, we have 7 years' experience of delivering this intervention and carrying out focus groups with mothers experiencing maternal mental health issues, as well as training in ethical practice and wider experience of carrying out clinical trials with those living with a range of mental health conditions. We have delivered training to the project partners facilitating these focus groups, based on our experience, and will also be available to discuss any issues that arise, should this be necessary. We have now added into the manuscript that this training will be delivered by UCL.
17	5. Given that the interventionists work with vulnerable people, please highlight the plans for supervision and securing the interventionists' wellbeing. And how are they trained to ensure that they understand the protocol and will be ready to comply to treatment fidelity measures.	We have a strong supervisory structure in place, whereby on-the-ground teams engage in bi-weekly online meetings with the strategic management team. Any issues arising are discussed and we ensure staff wellbeing, ethical practice and fidelity of delivery. This detail has now been added into the manuscript in the 'participants' section where we note the management structure of the project.
18	6. How are data managed? Will the team use systems such as Redcap or the like?	Thank you for noting this omission. We have now added a 'Data protection' section under 'Ethics and dissemination' where we include this detail.
19	The title says "scalability", but the aims state "feasibility study". The Abstract talks about group singing being "scaled up", which is unclear to me. So, what is it then? Otherwise scalability seems to play no role in the description.	Thank you for noting this. The study is to explore the feasibility of rolling out this intervention in different contexts, so in this sense we refer to it as a feasibility (of implementation) study. However, as the study is based on pre-existing research (see ref 24), we consider this intervention a 'scaling up' of the initial intervention as proposed by Fancourt & Perkins (2017). We test the feasibility of implementing an intervention,

		which itself is 'scaled up' from the foundations laid by researchers in the UK. As we have noted this complexity of scaling up a UK intervention and exploring the feasibility of this in a new context in the abstract and throughout, we have left the title unchanged.
20	"Although Postpartum depression (PPD) affects around one in seven women globally, there is currently no complete treatment solution." I am struggling with this line. What does it mean? Trying to be more constructive, I suggest to say: "Postpartum depression (PPD) affects around one in seven women globally. These women are in need for non-pharmaceutical treatment strategies." - Perhaps something like that would be more appropriate.	This amend has now been made.

VERSION 2 – REVIEW

REVIEWER	Melissa Forbes University of Southern Queensland, Centre for Heritage and Culture
REVIEW RETURNED	25-Oct-2022

GENERAL COMMENTS	Thank you for your revised manuscript which addresses the comments raised in my review. I wish you every success in this exciting and important project and look forward to reading the results in due course.
--

REVIEWER	Gunter Kreutz Carl von Ossietzky Universitat Oldenburg Institut fur Musik, Music
REVIEW RETURNED	13-Nov-2022

GENERAL COMMENTS	Thank you for addressing my points. Good luck!
--